



# The regional temperature implications of strong air quality measures

Borgar Aamaas[1], Terje K. Berntsen[1,2], Bjørn H. Samset[1]

[1]CICERO Center for International Climate Research, PB 1129 Blindern, 0318 Oslo, Norway
[2]Department of Geosciences, University of Oslo, PB 1047 Blindern, 0316 Oslo, Norway

*Correspondence to*: Borgar Aamaas (borgar.aamaas@cicero.oslo.no)

**Abstract.** Anthropogenic emissions of short-lived climate forcers (SLCFs) affect both air quality and climate. How much regional temperatures are affected by ambitious SLCF emission mitigation policies, is however still uncertain. We investigate the potential temperature implications of stringent air quality policies, by applying matrices of regional temperature responses to new pathways for future anthropogenic emissions of aerosols, methane ($CH_4$) and other short-lived

gases. These measures have only minor impact on $CO_2$ emissions. Two main options are explored, one with climate optimal reductions (i.e. constructed to yield a maximum global cooling) and one with maximum technically feasible reductions. The temperature response is calculated for four latitude response bands (90-28° S, 28° S-28° N, 28-60° N, and 60-90° N) by using existing regional temperature change potential (ARTP) values for four emission regions: Europe, East Asia, shipping, and the rest of the world. By 2050, we find that global surface temperature can be reduced by -0.3±0.08 °C with climate-

optimal mitigation of SLCFs relative to a baseline scenario, and as much as -0.7 °C in the Arctic. Cutting $CH_4$ and BC emissions contribute the most. This could offset warming equal to approximately 15 years of current global $CO_2$ emissions. If SLCFs are mitigated heavily, we find a net warming of about 0.1 °C, but when uncertainties are included a slight cooling is also possible. In the climate optimal scenario, the largest contributions to cooling comes from the energy, domestic, waste, and transportation sectors. In the maximum technically feasible mitigation scenario, emission changes from the sectors

industry, energy, and shipping will give warming. Some measures, such as in the sectors agriculture waste burning, domestic, transport, and industry, have outsized impact on the Arctic, especially by cutting BC emissions in winter in areas near the Arctic.

## 1 Introduction

Poor air quality is an issue of global concern, with health and welfare impacts affecting billions of people (WHO,

2016;Dockery et al., 1993;Di et al., 2017). Additionally, many of the components that make up air pollution also lead to radiative forcing impacting climate, through scattering or absorbing solar radiation, or by acting as greenhouse gases (Myhre et al., 2013b;von Schneidemesser et al., 2015). The net and individual climate impacts of present emissions of such short-lived climate forcers (SLCFs) have been extensively studied, but are however still poorly constrained (Stohl et al., 2015;Aamaas et al., 2016;Myhre et al., 2017;Samset et al., 2018).




In the coming decades, mitigation of $CO_2$ and other long-lived greenhouse gases (LLGHGs) is vital for the success of the goals in the Paris Agreement (UNEP, 2016). Concurrently, we expect large changes in SLCF emissions, in response to air quality policies, additional climate change mitigation efforts, and due to co-emissions with LLGHGs. As some SLCFs cool the climate, others warm it, and some may do both at different times after emission, the exact mitigation pathways of SLCFs will be of importance for the near-term rate and magnitude of warming – both globally and regionally. While several studies have analysed the impact on $CO_2$ mitigation of SLCFs (e.g., Rogelj et al., 2014), our study does not consider $CO_2$ emissions, but investigates a set of air quality measures that mainly influence SLCFs emissions.

Designing mitigation measures with both air quality and climate change in mind is however not straightforward, as warming SLCFs are often co-emitted with cooling SLCFs. Some authors have argued that a mitigation focus on SLCFs can be counterproductive, as this may lead to relaxing efforts on reducing $CO_2$ emissions (Pierrehumbert, 2014;Shoemaker et al., 2013). However, if this is done in a consistent way using emission metrics with appropriate time-horizons, this can be avoided (Berntsen et al., 2010). Another argument against SLCF mitigation today is that the long-term cooling potential of emission reductions is limited, and that delaying mitigation of SLCFs has only minor impact on temperature stabilization and peaking in the future (e.g., Pierrehumbert, 2014). However, SLCF mitigation is already occurring as part of air quality policy (Li et al., 2017) and is expected to continue in the coming decades regardless of the level of climate mitigation ambitions (Victor et al., 2015;Rao et al., 2017). Stohl et al. (2015) showed (applying the Absolute Regional Temperature change Potential (ARTP) methodology) that climate optimal reductions of SLCFs, i.e. the combination of measures which maximize temperature reduction, may lower the global temperature by 0.22 °C in 2041-2050. In comparison, a complete removal of anthropogenic aerosol emissions (BC, OC and $SO_2$) would induce a global mean surface heating of 0.5–1.1°C, according to four recent climate models (Samset et al., 2018). Going beyond temperature and precipitation impacts, SLCF emission mitigation is also known to have close links with a range of the UN Sustainable Development Goals (Haines et al., 2017).

Recently, Stohl et al. (2015) gave a general overview of the temperature reduction potential. Our study explores these findings for individual emission regions and emission sectors using updated data and methods, following pathways that focus on air quality concerns. Our focus is on the temperature effects of SLCFs, however, mitigation of these components can also help to achieve several of the Sustainable Development Goals (Shindell et al., 2017).

A detailed look into what sectors and regions that contribute to the mitigation potential from SLCF reductions requires a comprehensive emission dataset. As part of the ECLIPSE project, emission inventories and scenarios for future emissions (for the period 1990-2050) of SLCFs were produced (Klimont et al., 2017;Klimont et al., In prep.). The scenarios describe three different futures with different mitigating ambitions. The baseline scenario assumes implementation of current legislation (CLE). Both current and planned environmental laws are included while considering known delays, but assuming full enforcement in the future (Stohl et al., 2015). The most ambitious mitigation scenario is labelled "the maximum





technical feasible reductions" (MTFR), where SLCFs are cut as much as possible (although without changes in consumer behaviour, structural changes in transport, agriculture or energy supply or additional climate policies) due to air quality concerns. This is a very policy demanding scenario, as most emissions are reduced by 60-80 % within a few decades. However, similar trends have historically been seen for emissions of $SO_2$ and $NO_x$ in Western Europe and North America

(Amann et al., 2013;Rafaj et al., 2015). The third scenario can be seen as a subset of MTFR, as this climate-optimal mitigation scenario (SLCP$_{scen}$) includes roughly 50 different mitigation measures on SLCFs from the MTFR catalogue of measures that avoid warming. The scenario name SLCP$_{scen}$, based on the scenario name SLCP given by IIASA, should not be mixed up with the term short-lived climate pollutant. Only measures that are estimated to lead to net global cooling, while reduction of co-emitted cooling species are accounted for, are included. Every selected measure gives a net cooling based on

the Global Temperature change Potential values for a time horizon of 20 years, given a linear ramp-up of emission measures over a time period of 15 years (Stohl et al., 2015). These mitigation measures can be grouped into three categories of measures. First, measures on emissions of $CH_4$ that can be centrally implemented (e.g., recovery and use of gas from oil and gas industry). Second, technical measures on BC emissions from small stationary and mobile sources (e.g., eliminating high-emitting vehicles). Third, non-technical measures to eliminate emissions of BC (e.g., banning of open-field burning of

agricultural residues). The 17 largest mitigation measures that contribute to more than 80 % of the climate benefit are given in Table 3 in Stohl et al. (2015). Stohl et al. (2015) showed that these measures have only minor impact on emissions of $CO_2$.

The global and regional temperature impact of these emission scenarios should ideally be calculated with the most advanced Earth System models, but can be approximated and explored quickly for different emission components and sectors with

emission metrics. Perturbation in the global temperature is most commonly calculated with the absolute Global Temperature change Potential (AGTP) (Shine et al., 2005;Shine et al., 2007), while the regional temperature distribution in broad latitude bands can be investigated with absolute Regional Temperature change Potential (ARTP) (Shindell and Faluvegi, 2010). AGTP and ARTP quantify the warming per unit emission and can be seen as building blocks to analyse different emission scenarios. As described by Aamaas et al. (2017), the temporal regional temperature response of any emission scenario or

difference between scenarios can be calculated with a convolution given the emission dataset and ARTP values.

In this study, we use mitigation datasets of SLCFs and regional temperature metrics to calculate the potential of SLCF mitigation for reducing global and regional temperatures. Our analysis builds on Stohl et al. (2015), while the novelty of our work is that we estimate the temperature change potentials of mitigating different emission regions and emission sectors. We

investigate what species can contribute the most to spatially and temporally resolved mitigation. The methods are described in Sect. 2. The results are presented in Sect. 3 and discussed in Sect. 4. We conclude in Sect. 5.



## 2 Methods

### 2.1 The ECLIPSE dataset

The analysis in this paper is based on emission pathways from the ECLIPSE emission project (Klimont et al., 2017;Klimont et al., In prep.). Briefly, ECLIPSE estimated possible future emission values based on different ambition levels for mitigation

of SLCFs. The emission pathways we use are shown in Fig. 1. Emissions are given for seven SLCFs: Black carbon (BC), organic carbon (OC), sulphur dioxide ($SO_2$), nitrogen oxides ($NO_x$), carbon monoxide (CO), volatile organic compounds (VOC), and methane ($CH_4$). The sectors included are agriculture (agr), agriculture waste burning (awb), domestic (dom), energy (ene), industry (ind), solvent (slv), transportation (tra), waste (wst), and shipping (shp). The datasets contain information on the seasonal cycle, such as larger emissions from residential heating and cooking in winter (not shown).

### 2.2 SLCF mitigation pathways

As we are interested in how much mitigation of SLCFs can contribute towards reducing global and regional temperatures in the next decades, we construct two mitigation datasets from these pathways. The first is the emission difference between the mitigation scenario SLCP_scen (see Fig. 1B) relative to the baseline CLE (see Fig. 1A). The second is the emission difference between the mitigation scenario MTFR (Fig. 1C) relative to CLE. As the MTFR dataset in the ECLIPSE project is only

given for the 2030-2050 period, we assume a linear trend between 2015 and 2030 for MTFR. We use the most recent version of the datasets, ECLIPSE V5a (Klimont and Heyes, 2016). The ECLIPSE dataset is very detailed. Here, we aggregate regionally and seasonally as necessary to match the format of the ARTP values available (Aamaas et al., 2017). We interpolate linearly between the emissions points, which are given every five years and in some cases every ten or 20 years. Since the emission scenarios from ECLIPSE go until 2050, we keep emission levels constant between 2050 and 2100, as we

are not aware of any scenarios that are compatible with the ECLIPSE scenarios and contain the level of detail needed for our analysis. A large share of the emissions are also mitigated by 2050; hence, the temperature potential of further emission cuts after 2050 is limited.

### 2.3 Regional temperature potentials (ARTP)

The ARTP values applied come from the study by Aamaas et al. (2017). They give values for each species for emissions

occurring in Europe (EUR), East Asia (EAS), global shipping (SHP), and the rest of the World (ROW), as well as separating between Northern Hemisphere summer and winter emissions. The temperature response is given for four latitude response bands (90-28° S, 28° S-28° N, 28-60° N, and 60-90° N). The temperature response in latitude band l in year t is given by a convolution:

$$\Delta T_{i,r,s,l}(t) = \sum_{u=1}^{2} \int_0^t \Delta E_{i,r,s,u}(t') \times ARTP_{i,r,s,u,l}(t - t')dt' \tag{1}$$





for species i emitted in region r from emission sector s during season u (the year is divided into two seasons, summer u=1 and winter u=2). The emission difference between a mitigation scenario and the reference scenario is ΔE. As the final year in the ECLIPSE emission scenarios is 2050, our main case is the temperature impact in 2050.

Uncertainties (1 standard deviation) in the global temperature response have been estimated given a Monte Carlo analysis of 100 000 simulations. This analysis is based on a probability density function defined by model based estimates of uncertainties in direct radiative forcing from the literature (Myhre et al., 2013b;Myhre et al., 2013a) with the same treatment of radiative forcing uncertainty as in Lund et al. (2017) (see also the Supplement). Radiative forcing from each species is treated as a random variable. The distribution for the total uncertainty is derived by summing the probability density

functions of all species. We assume that the radiative forcing uncertainties are independent in these calculations. Also, note that the multi-model studies used as input were run with unified emissions. This particularly affects BC, where the current substantial uncertainty in annual emissions (Bond et al., 2013;Cohen and Wang, 2014) will not be represented. We compare our derived uncertainties to the influence of low and high climate sensitivities in the literature, 1.5 and 4.5 °C for a doubling of $CO_2$ (Bindoff et al., 2013). Here, we adopt a lognormal distribution and assume the value range covers 1 standard

deviation. Uncertainties are not given for the latitude bands as a formal quantification of uncertainties for the ARTPs has not been produced.

## 3 Results

If SLCFs are mitigated in a climate-optimal manner, we estimate a maximum change in global temperature of -0.33±0.083 °C by 2050, relative to current legislation, increasing to about -0.4 °C later in the century ending at -0.44±0.11 °C in 2100

(see Fig. 2A, black line). The global temperature change is calculated as the area-weighted sum of the net regional changes given by equation 1. The temperature response of aggressive mitigation of SLCFs (MTFR) leads gradually to a small net warming of 0.059±0.15 °C in 2050 relative to the baseline, which seems to be counter-productive in terms of goals limiting the global temperature increase (see Fig. 2B). As the uncertainty interval is large, since large emission cuts of warming and cooling components cancel each other almost out (about 0.7 °C cooling and warming in 2050, see Fig. S3 in the Supplement

for cooling and warming separated for both MTFR and SLCP$_{scen}$), we cannot rule out that this scenario may lead to cooling. In the climate-optimal scenario (SLCP$_{scen}$), CH$_4$ (-0.21±0.021 °C in 2050 and increasing in magnitude) and BC (-0.19±0.073 °C in 2050) are the main drivers of the temperature reductions. The measures will also reduce co-emissions of cooling species causing a warming of more than 0.2 °C in 2050. The main warming contributions are emission reductions of OC and NO$_x$, with small impacts from other SLCFs. The main difference to the maximum reduction scenario (MTFR) is the large

warming contribution for MTFR (0.35±0.12 °C in 2050) from SO$_2$ reductions, as well as additional warming from NO$_x$ reductions.



The Arctic (60-90° N) is the region that is the most sensitive to the mitigation scenarios (see Fig. 3), followed by northern mid-latitudes (28-60° N), as the climate sensitivities are largest for those regions and most of the emissions occur on the Northern Hemisphere (Aamaas et al., 2017). In SLCP$_{scen}$, the cooling in the Arctic (-0.69 °C in 2050) is more than twice the global average. This sensitivity in the Arctic is larger than for reductions of $CO_2$, which would be roughly 50% when

applying the ARTP concept on $CO_2$. This amplification in the Arctic is larger than the average for mitigation of European emissions and smaller for mitigation of East Asian emissions. Measures on BC emissions during winter in the Northern Hemisphere contribute to this amplification. In terms of sectors, mitigation measures on SLCFs from agriculture waste burning, domestic, transportation, and industry have larger than average influence on the Arctic relative to the global average (Fig. 4). Some variability is also seen for the Arctic. While MTFR will lead to warming globally relative to CLE, a cooling

of the same magnitude is estimated for the Arctic. The net cooling in the Arctic is driven by emissions from rest of the World, while mitigation in the shipping sector leads to warming for both and the net effect of European mitigation is near zero (see Fig. 3b).

The emission region that contributes the most in the mitigation scenarios is Rest of the World. In the Supplement, we

indicate that rest of Asia and other developing regions are the most important regions (as seen in Stohl et al., 2015), although our ARTP dataset limits us from making clear conclusions of what sub regions have the largest cooling potential. In SLCP$_{scen}$, mitigation leads to cooling from all emission regions and emission sectors except global shipping. In MTFR, warming globally is estimated for rest of the world and shipping, while near zero change for Europe and a cooling contribution for East Asia.


The emission sectors that give the largest cooling in SLCP$_{scen}$ are energy, domestic, waste, and transportation (see Fig. 4b). In the Arctic, the order changes with domestic becoming the most important sector and transportation moved up to third place, mainly due to the large warming of BC in the Arctic. Shipping is the only sector that causes a small warming when mitigated. While MTFR lead to a net warming, only three out of the nine sectors contribute to that, the sectors industry,

energy, and shipping. Even for the energy sector, mitigation in East Asia leads to cooling. Most of the mitigation measures found unsuitable in a climate-optimal scenario can be placed in those sectors.

In Fig. 5, the uncertainties for the global temperature responses in Figs. 3 and 4 based on uncertainties in radiative forcing are compared to the uncertainty given different climate sensitivities. The uncertainties for the radiative forcing give generally

a larger span than the climate sensitivity when a broad mix of emissions are mitigated, such as in the MTFR. For individual components, the range in climate sensitivities leads to a larger span than uncertainty in radiative forcing.



## 4 Discussion

The method applied here (Sect. 2.3) estimate the long-term response to a sustained change in SLCF emissions. However, in the current climate (here 2015), the climate has not reached the full response of sustained SLCF emissions at the current level due to the thermal inertia of the system. We have also estimated the temperature perturbations after 2015 running a

transient simulation through 2015 using also historic emissions of SLCFs and applying the same methodology as in Sect. 2.3. The potential for temperature reductions is reduced by up to 0.04 °C in 2050 and 0.05 °C in 2100 when this masked warming is included.  Hence, the actual global temperature reduction is -0.30 °C by 2050 in SLCP$_{scen}$, when climate variability is excluded.

For the mean of the 2041-2050 period, our estimate of global temperature change of -0.29 °C relative to the baseline is higher than -0.22 °C calculated by Stohl et al. (2015), which may be due to the usage of different versions of the ECLIPSE emission datasets, as well as some updates in the ARTP values.

A temperature change in 2050 of -0.33 °C relative to the baseline could potentially offset a large increase in $CO_2$ emissions.
If we weight with ARTP with a time horizon of 30 years, approximately the number of years until 2050, this temperature change is the same as about 520 Gt $CO_2$, or 15 years of current global $CO_2$ emissions. A climate optimal mitigation of SLCFs can therefore contribute to limiting the global temperature increase; however, only in addition to sustained $CO_2$ mitigation (e.g., Shoemaker et al., 2013).

SLCFs are mitigated due to different concerns, including that it contributes to achieving several of the Sustainable Development Goals (Shindell et al., 2017). Hence, while a climate-optimal mitigation strategy on SLCFs may be needed, in addition to reducing $CO_2$ emissions, to contribute in avoiding global warming above the temperature targets in the Paris Agreement, measures undertaken to reduce air pollution and other problems are likely to lead to higher levels of warming. In this respect, climate-optimized mitigation of SLCFs can be considered as a type of geoengineering, as we keep emitting
cooling substances. This is not an obvious nor trivial choice due to the higher levels of air pollution it entails, and would likely meet political resistance, as the ability to also address air pollution is seen as a main motivation for SLCF mitigation (Victor et al., 2015), although the two problems are viewed as interlinked (Tvinnereim et al., 2017). Thus, the feasibility of only executing the climate-optimal measures is lower than if there were no other concerns. SLCFs mitigation will lead to numerous other benefits, reducing health problems, increasing yields from agriculture, and achieving several of the
sustainable development targets (UN, 2015;Haines et al., 2017). Many of the measures with the largest overall economic benefits involve $SO_2$ reductions, measures that may be difficult by policymakers to neglect while prioritizing less beneficial measures that are climate-optimal. Another issue is the choice of baseline for evaluation of temperature change. We apply here the most recent ECLIPSE emission dataset from July 2015, while measures taken and planned legislation after that date



will, in particular, lower $SO_2$ emissions. The two main consequences are that the warming impact of MTFR is probably smaller or non-existent, and that limiting the global temperature increase to 1.5 °C is harder as more $SO_2$ emissions are removed than in a climate-optimal SLCF mitigation scenario.

SLCFs are also co-emitted with $CO_2$. The ECLIPSE mitigation dataset makes use of external projections of energy use and industrial production and does not include mitigation measures directly on $CO_2$. Stohl et al. (2015) argue that the measures included in this study have no significant impact on $CO_2$ emissions. However, Rogelj et al. (2014) showed that mitigation of $CO_2$ will lead to reductions of SLCFs. Hence, the potential cooling effect of dedicated reductions in emissions of warming SLCFs may be limited by successful mitigation of $CO_2$. As global temperature may peak or stabilize some time after 2050,

the temperature reduction by mitigating SLCFs can be seen as more critical at reducing this peak or level than reducing global temperature in 2050, the year we focus on in this study.

While the calculations here could also be based on AGTP values, Aamaas et al. (2017) argue that the regionality and seasonality included in the emission dataset and in the metric value give added values. Regional responses, such as the

higher efficacy in the Arctic due to emissions close to the Arctic is better captured than global averages. Users of these results may also find estimated temperature responses in latitude bands more interesting than a global average. While previous studies have used ARTP values to calculate the temperature impact of SLCF mitigation globally (Stohl et al., 2015) and in the Arctic (Sand et al., 2016), we also show the temperature impact in the regions where most people live, such as in the 28-60° N latitude band. For this band, the net temperature reduction in 2050 in the SLCP$_{scen}$ scenario relative to the

baseline is 0.48 °C, or almost 50% larger than the global average.

Emission metrics are based on the current atmospheric composition and linearity, hence, an 80 % reduction of a pollutant is assumed to give twice the impact of a 40 % reduction. While this holds for small perturbations, this assumption may be inaccurate for the large SLCFs reduction by 2050 in the SLCP$_{scen}$ and MTFR scenarios. Chen et al. (2018) recently

quantified the uncertainties by assuming linearity and found an error up to 15% for the direct radiative forcing efficiency for BC and OC, when assuming a total phaseout of emissions. The uncertainties can be larger for the indirect radiative forcing, especially in high emitting regions. Another assumption is the choice of constant emissions after 2050. This was chosen, as we were unable to combine with other scenarios with emission data after 2050, while a reduction of emissions to varying degree in all three scenarios may occur after 2050. Newer studies (e.g., Stjern et al., 2017;Baker et al., 2015) have also

shown that the warming of BC emissions is smaller than implicit included with the emission metric values used here; hence, the cooling potential of reducing BC emissions is likely smaller than estimated by us. However, our dataset is in the lower end of the range given by Samset et al. (2018) (0.5–1.1°C for removing all anthropogenic emissions of BC, OC, and $SO_2$) and thus not outside of the likely range given by state-of-the-art knowledge.




Different stakeholders may be interested in different aspects of our calculations. Decision makers can easily combine their own emission datasets with ARTP values to investigate what is most relevant for them. As the dimensions are many, we present additional figures in the Supplement, such as how the regional temperature change is for different times throughout the 21st century.

**5 Conclusion**

This study has not analysed scenarios with $CO_2$ mitigation or measures on SLCFs that will also result in emission cuts of $CO_2$. However, we have estimated the temperature effects of different air quality measures on SLCFs emissions. We have shown that mitigation of SLCFs can contribute to reduce the global and regional temperatures in the next few decades, if mitigation is optimized with regards to temperature change. On the other hand, mitigation of SLCFs to gain other benefits

can be counter-productive for limiting the temperature increase, especially if we cut emissions of $SO_2$. A global temperature reduction from SLCF mitigation of about -0.4±0.1 °C is technically feasible in the second part of the 21st century. Emission reductions of $CH_4$ and BC will contribute the most. The sectors with the largest shares contributing to cooling are energy, domestic, waste, and transportation in the SLCP$_{scen}$ scenario, while aggressive emission cuts will lead to warming from industry, energy, and shipping. The net response in the SLCP$_{scen}$ scenario is almost 50% larger than the global average for

the 28-60° N latitude band and more than the double for the Arctic. BC emissions drives this as BC emissions during winter in the Northern Hemisphere will have much larger contribution than when looking at global and annual averages. The Arctic is the most influenced by mitigation in the sectors domestic, energy, and transportation. The feasible temperature reductions may be smaller due to several reasons, such as the entangling of SLCFs and $CO_2$ emissions, the unlikely option by policymakers of leaving out measures that are highly beneficial for health that are not climate-optimal, and newer studies

indicating a smaller temperature impact of BC emissions.

*Data availability.* The analysis is based on two datasets. The ECLIPSE emission data can be downloaded from http://www.iiasa.ac.at/web/home/research/researchPrograms/air/ECLIPSEv5a.html. The ARTP values applied can be found in Aamaas et al. (2017).


*Author contribution.* TKB and BA developed the idea of the study. BA compiled the needed data and modelled the regional temperature responses. BA lead the writing of the paper, with contributions from TKB and BHS.

*Competing interests.* The authors declare that they have no conflict of interest.






*Acknowledgements.* This work was supported by the Research Council of Norway within the project "Kortlevede klimadrivere i en 1.5 graders verden" (project no. 261728). We thank Zbigniew Klimont for help with the ECLIPSE emission dataset and Steffen Kallbekken for commenting a draft version of this article.

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

**Figure 1: The global emission levels relative to the 1990 level for CLE (A), SLCP$_{scen}$ (B) and MTFR (C). The 1990 emission level for each SLCF is normalized to 100.**

10 **Figure 2: Global temperature response due to the SLCP$_{scen}$ (A) and MTFR (B) scenarios relative to the baseline CLE scenario. Future global temperature change will also be impacted by historic and baseline emissions, which are not accounted for here.**

**Figure 3: The temperature response in the latitude bands and globally in 2050 for emission regions and emission sectors for SLCP$_{scen}$ (A) and MTFR (B) scenarios relative to the baseline CLE. The emission regions are Europe (EUR), East Asia (EAS), global shipping (SHP), and the rest of the World (ROW). The emission sectors are agriculture (agr), agriculture waste burning (awb), domestic (dom), energy (ene), industry (ind), solvent (slv), transportation (tra), waste (wst), and shipping (shp).**

**Figure 4: The temperature response in the latitude bands and globally in 2050 for emission sectors and species for SLCP$_{scen}$ (A) and MTFR scenario (B) relative to the baseline CLE. Future global temperature change will also be impacted by historic and baseline emissions, which is not accounted for here. The emission sectors are agriculture (agr), agriculture waste burning (awb), domestic (dom), energy (ene), industry (ind), solvent (slv), transportation (tra), waste (wst), and shipping (shp).**

20 **Figure 5: The global temperature response in 2050 in mitigation scenarios relative to the baseline for emission regions and emission sectors for SLCP$_{scen}$ and MTFR scenario. 1 standard deviation uncertainties are included. The blue (black) error bars indicate the 1 standard deviation in the RFs based on no inter-species correlation in the uncertainties for emission regions (emission sectors). The grey error bars show the uncertainty in the climate sensitivity.**



Figure 1



Figure 2

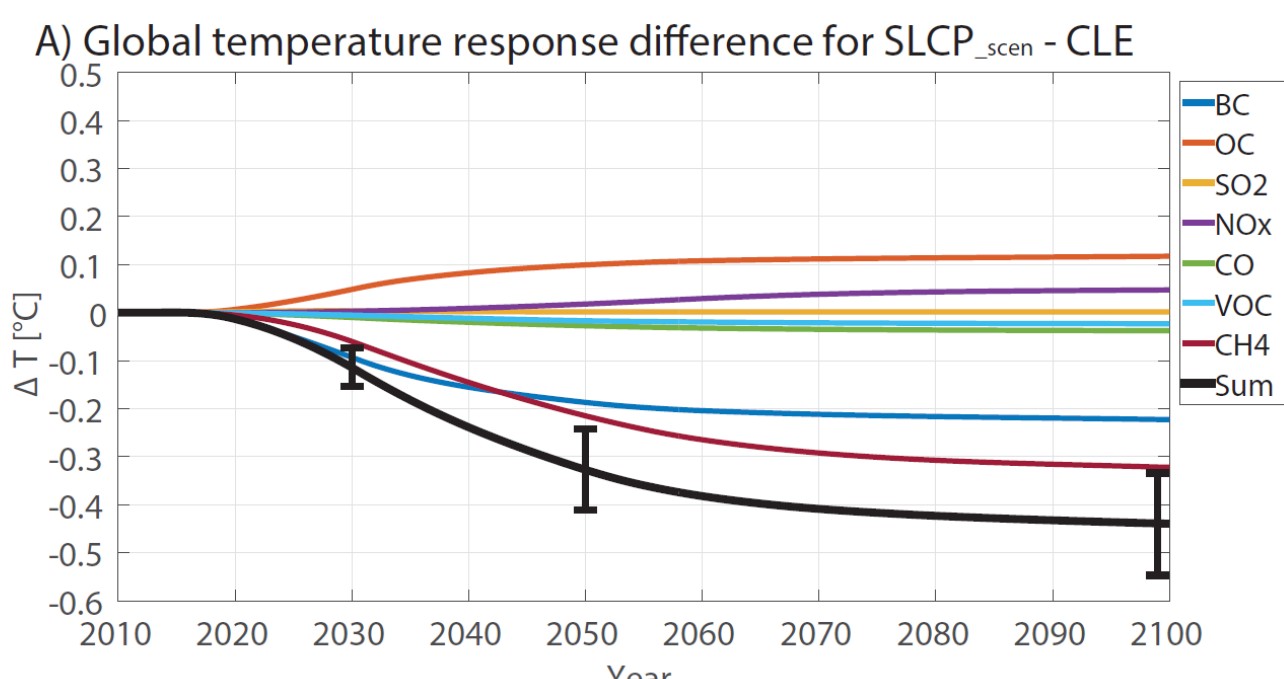

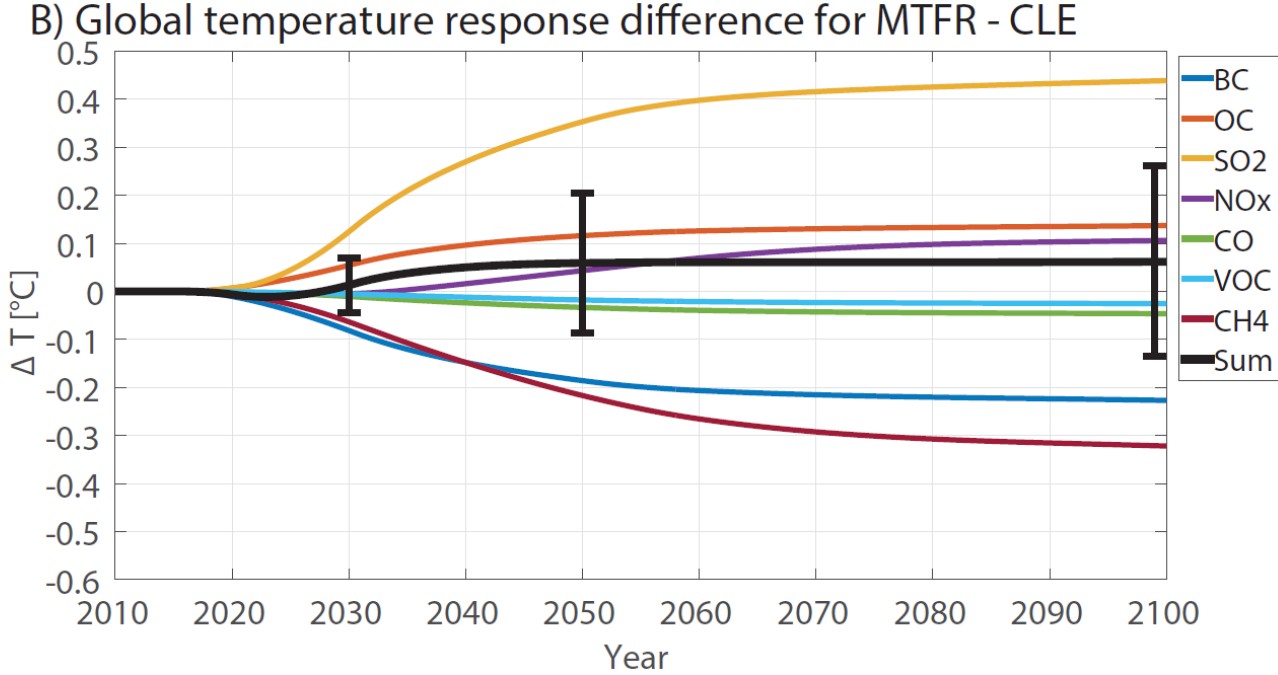



Figure 3

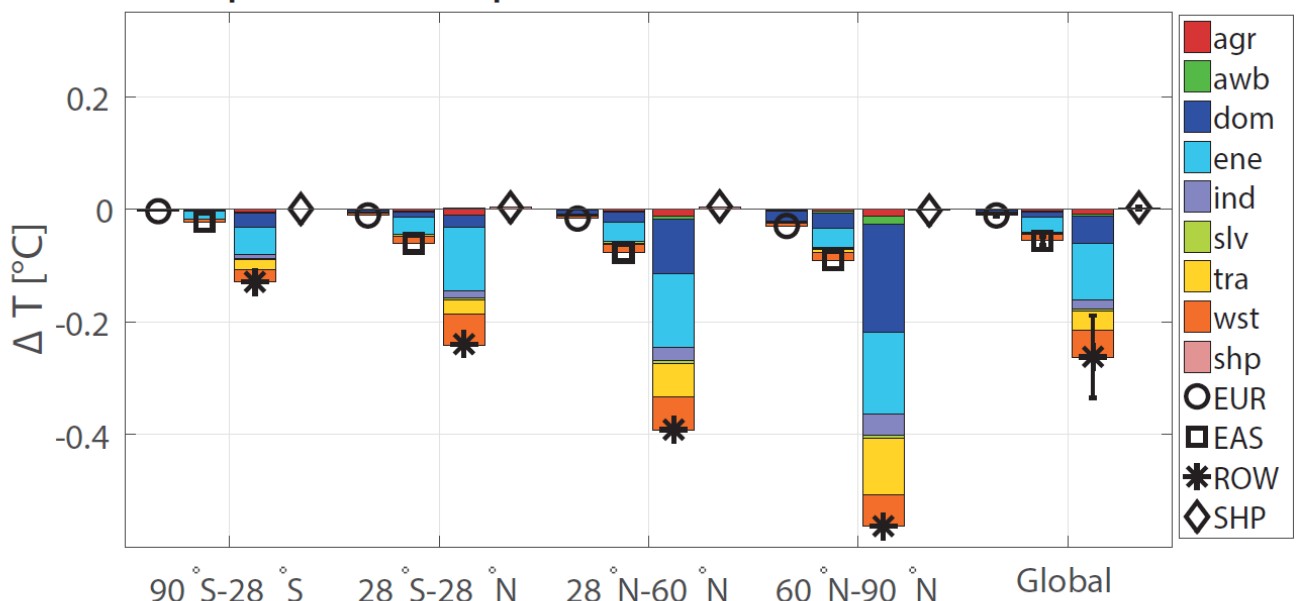

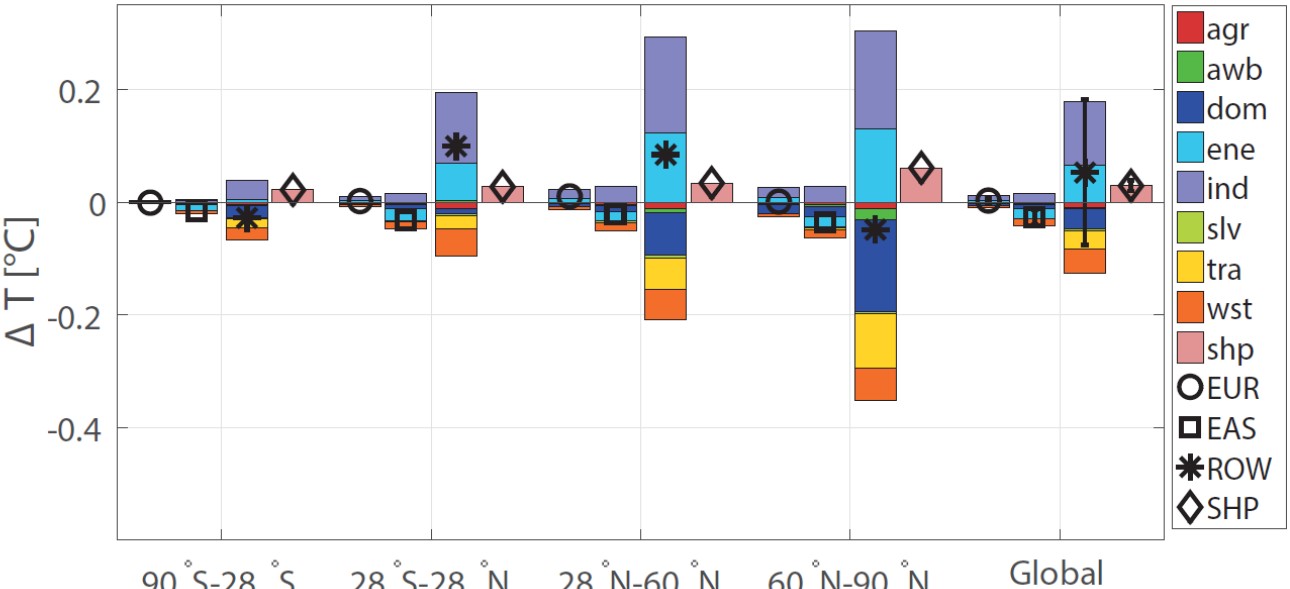





Figure 4



Figure 5

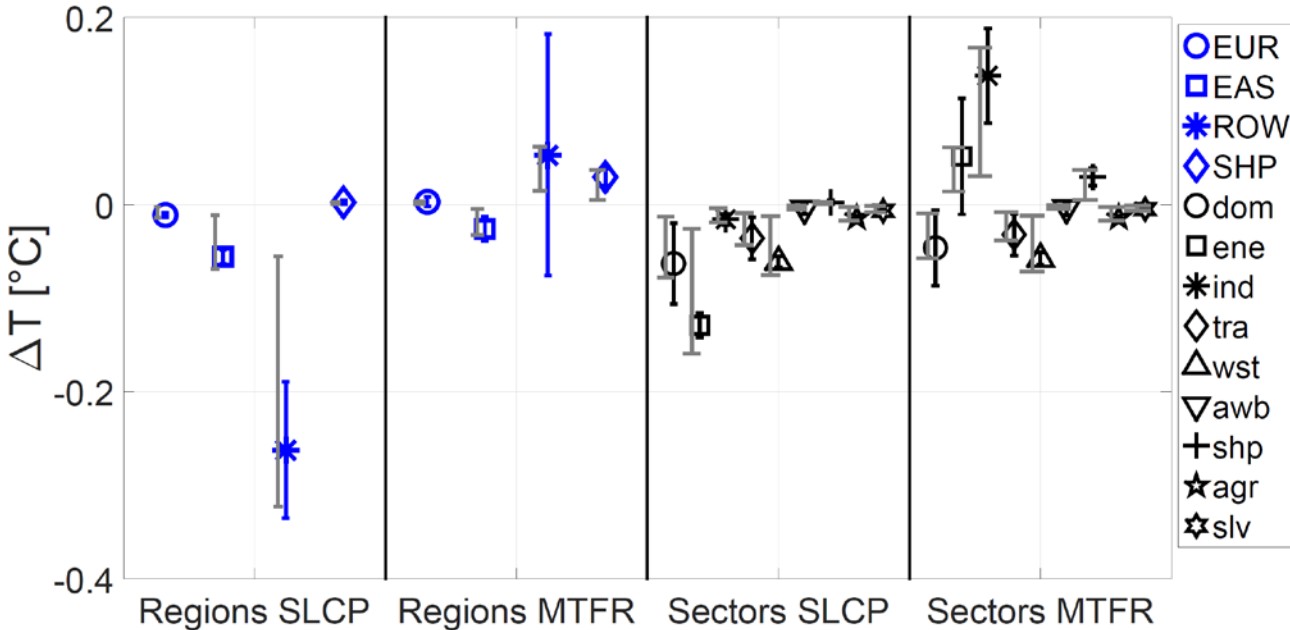