# Peer review of "The regional temperature implications of strong air quality measures"

_Atmospheric Chemistry and Physics, 2019_

## Referee Comment (RC1) · Anonymous Referee #2 · 24 Aug 2019

In this study, the authors investigate the potential temperature implications of stringent air quality policies, by applying matrices of regional temperature responses to new pathways for future anthropogenic emissions of aerosols, methane ($CH_4$) and other short-lived gases. This is an interesting and relevant topic since there are still a lot of uncertainties on how regional temperatures are affected by ambitious SLCF emission mitigation policies.

**General comments**

The Introduction is too short, I suggest the authors to add more information about SLCF description. For example, here you show results for BC, OC, $SO_2$, NOx, CO, VOC, and $CH_4$. Some description about their cooling/warming impact of them will help to a better understand of the results. Also, maybe a bit more description of the ECLIPSE project would be good, since this works is strongly connected to it.

Some figures are not well described in the text, for example, the authors directly mention Fig. 2 or 3 after describing a result obtained. The figures should be defined saying what it is representing there, and if needed, some explanation about how to interpret the graphic (if I got it right, the different symbols in fig 3 shows the influence of the different sectors of that region in a latitude band). This way it will be easier to follow the text. Please, do it whenever the figure has not been described in advanced. There is also a lack of mention some figures that I will point out in the technical comments.

A better organization of the results must be done. It would be better to organize them in subsections, like "Global/Regional temperature change" or/and "Results by regions", as an example. Furthermore, more quantitative results could be added. Complementing with a table would be helpful for a better overview of the results found in this study (and comparing the results found in Stohl et al., 2015).

**Specific comments**

Page 1, line 16, authors state that "*cutting CH4 and BC emissions contribute the most. This could offset warming equal to approximately 15 years of current global CO2 emissions.*" How do you get to this conclusion? I haven't seen it in the manuscript.

Page 2, line 24, when mentioning the work of Stohl et al., 2015, although the authors mention it throughout the text, it will be good to have more description of the what they found.

Page 2, line 29, here the description of the scenarios is done. I have several comments:

- The description of the SLCP_scen is not clear for me, so the difference is that the SLCP_scen only has 50 different mitigation measures on SLCFs compared to the

MTFR; a) how many has the MTRF scenario? And b) in what is it based to be called optimal?

- It would be helpful to have the scenarios description as a list or as a table with a short description.
- Check that you call the baseline scenario as "baseline" in the text. There are couple of times where you call it CLP, and sometimes is confusing to follow all the acronyms.

Page 3, line 7, IIASA has not been described.

Page 5, line 5, do you refer to the results shown in figure 5? If so, you could refer to it here.

Page 5, line 20, "The global temperature change is calculated as the area-weighted sum of the net regional changes given by equation 1." Can be added to the subsection 2.2, after describing the ARTP.

Page 5, line 28, "a warming of more tan 0.2ºC" how do you get this value?

Page 6, what about the warming temperature response found in fig. 3b and 4b? Authors only focus on cooling temperature response results.

Page 7, line 14, to what scenario does the value -0.33 ºC correspond?

**Technical comments**

Page 1, line 13, "using existing regional temperature change potential (ARTP)" did you mean, *using **absolute** existing regional temperature change potential?*

Page 1, line 25, add a comma after "pollution".

Page 3, line 16, move "in Table 3 in Stohl et al. (2015) " to the beginning of the sentence in line 15 to avoid "Stohl et al. (2015). Stohl et al. (2015) "

Page 4, line 14, "CLE" to "baseline".

Page 5, line 25, add Fig. 2A somewhere in this line.

Page 5, line 30, add Fig. 2B somewhere in this line.

Page 6, line 9, "CLE" to "baseline".

Page 6, line 14, in "mitigation scenarios" do the authors refer to MTRF?. If so:

    Page 6, line 14, add Fig. 3b after "rest of the World".

Page 6, line 21, it should be Fig. 4a.

Page 6, line 25, add Fig. 3b after "to cooling.".

Page 10, line 11, doi is missing.

---

## Referee Comment (RC2) · Anonymous Referee #1 · 6 Sep 2019

General Comments

This is a well written and interesting study that explores how optimal mitigation of short-lived climate forcers could contribute to cooling (or reduced warming) over the next few decades. It provides useful information for policymakers considering how to limit warming through non-CO2 measures. There are, of course, many uncertainties, but this study demonstrates very nicely what could be done. My main concern is that the uncertainties in the ARTP values should be discussed more deeply than they are currently; this also extends to better explanations of the error bars on some of the figures (see below). If this point and the other, relatively minor points below are addressed, then I am happy to recommend publication in ACP.

Specific Comments

[Figure]

In the Abstract, clarify that mitigation of some SLCFs (e.g. SO2) leads to warming.

P1 l7 Change text to: '. . .policies is, however, still uncertain.'

P1 l21 'outsized impact' -> large impacts?

P2 l7 SLCFs -> SLCF

P2 l19 I think you need to say something like ". . .may lower the global temperature by 0.22C in 20141-2050 compared to a reference scenario."

P2 l19 "complete removal of anthropogenic aerosol emissions (BC, OC, SO2). . ." SO2 is, of course, an aerosol precursor, not an aerosol. "Complete" seems excessive, as I don't think you mean removal of species like NOx and NH3 (which are also aerosol precursors), so I would use slightly less all-encompassing language here.

P2 l22 ". . .a range of the UN SDGs." This is a bit vague – presumably you mean air quality, food security, etc. Can you be a little more specific?

P2 l24 potential of SLCFs.

P2 l29 delete 'that'

P2 l32 State year for 'current legislation' – 2019?

P3 l1 technically

P3 l20, l22 capitalise Absolute

P3 l25 How well constrained/model dependent are the (crucial) ARTP values? This is rather important and deserves some discussion. For example, is nitrate aerosol included in the model(s)? How are the indirect effects of aerosols handled in the model(s)? Do the models include interactive vegetation, e.g., that responds to air pollution induced damage? How do the models represent the mixing state of aerosols? Do we have any idea about how these missing processes (I am assuming they are missing) will affect the model results? I appreciate that you can only use state-of-theart models to make your best estimate of temperature responses, but some discussion of how uncertain the results (ie ARTP values) are should be included, to give some perspective. I note you do quote errors on your values – but I think these cover just the known unknowns.

P5 l18-l26 Please quote values +/- errors correctly. It is incorrect to quote -0.33 +/- 0.083 C. The error should probably only be quoted to one significant figure, although you may feel justified to quote to two, as you have done. But the value then needs to be quoted to the same number of decimal places as the error, i.e. it should be, e.g., -0.335 +/- 0.083 C, or -0.33 +/- 0.08 C. The same inconsistency appears on several of the subsequent lines.

P6 l2 on -> in

P6 l5 on -> to

P7 l2 estimates

P7 l11 usage -> use

P7 l20 it contributes -> they contribute

P8 l14 values -> value

P8 l15 is -> are

P8 l24 SLCFs reduction -> SLCF reductions

P8 l30 implicitly

P9 l3 how -> what

P9 l18 . . .may be smaller than those estimated here. . .

Figure 2 caption should explain the origin of the error bars.

Figure 4 caption should explain the origin of the error bars (on the global values).

Figure 5 caption – the explanation of the error bars could be clearer.

[Figure]

---

## Author Response (AR1)

We thank for the constructive and helpful comments from the Reviewers. Our answers are given in red. When we have accepted the suggestion, this is given by an "OK".

Reviewer 1:

General Comments

This is a well written and interesting study that explores how optimal mitigation of short-lived climate forcers could contribute to cooling (or reduced warming) over the next few decades. It provides useful information for policymakers considering how to limit warming through non-CO2 measures. There are, of course, many uncertainties, but this study demonstrates very nicely what could be done. My main concern is that the uncertainties in the ARTP values should be discussed more deeply than they are currently; this also extends to better explanations of the error bars on some of the figures (see below). If this point and the other, relatively minor points below are addressed, then I am happy to recommend publication in ACP.

We have added more details on the ARTP values, such as what processes are included, how they were estimated in the studies we refer to, as well as on uncertainties. We have also edited the explanations of the error bars. See specific replies below.

Specific Comments

In the Abstract, clarify that mitigation of some SLCFs (e.g. SO2) leads to warming. We have added this sentence to the abstract: "On the other hand, mitigation of other SLCFS (e.g., SO2) leads to warming."

P1 l7 Change text to: '…policies is, however, still uncertain.' OK

P1 l21 'outsized impact' -> large impacts? OK

P2 l7 SLCFs -> SLCF OK

P2 l19 I think you need to say something like "…may lower the global temperature by 0.22C in 2041-2050 compared to a reference scenario." OK

P2 l19 "complete removal of anthropogenic aerosol emissions (BC, OC, SO2)…" SO2 is, of course, an aerosol precursor, not an aerosol. "Complete" seems excessive, as I don't think you mean removal of species like NOx and NH3 (which are also aerosol precursors), so I would use slightly less all-encompassing language here. We have revised the sentence to:
"In comparison, a complete removal of anthropogenic emissions of black carbon (BC), organic carbon (OC) and SO2 (sulphate aerosol precursor) would induce a global mean surface heating of 0.5–1.1°C, according to four recent climate models (Samset et al., 2018)."

P2 l22 "…a range of the UN SDGs." This is a bit vague – presumably you mean air quality, food security, etc. Can you be a little more specific? We have reworded and extended this part to:
"Going beyond temperature and precipitation impacts, SLCF emission mitigation is also known to have multiple co-benefits and trade-offs with the UN Sustainable Development Goals (Haines et al., 2017). The co-benefits are generally larger than the trade-offs. Among the most well-known co-benefits, we find that SLCF mitigation will reduce air pollution and, hence, reduce premature deaths (SDG3), as well as reduce crop losses (SDG2)."

P2 l24 potential of SLCFs. OK

P2 l29 delete 'that' OK

P2 l32 State year for 'current legislation' – 2019? The year for current is 2015 in the emission dataset, so we added "(2015)" in the sentence.

P3 l1 technically OK

P3 l20, l22 capitalise Absolute OK

P3 l25 How well constrained/model dependent are the (crucial) ARTP values? This is rather important and deserves some discussion. For example, is nitrate aerosol included in the model(s)? How are the indirect effects of aerosols handled in the model(s)? Do the models include interactive vegetation, e.g., that responds to air pollution induced damage? How do the models represent the mixing state of aerosols? Do we have any idea about how these missing processes (I am assuming they are missing) will affect the model results? I appreciate that you can only use state-of-the-art models to make your best estimate of temperature responses, but some discussion of how uncertain the results (ie ARTP values) are should be included, to give some perspective. I note you do quote errors on your values – but I think these cover just the known unknowns. We have added a couple of paragraphs in the method section (Section 2.3) on the ARTP values and what processes they include.

"The ARTP dataset utilized here are presented in detail by Aamaas et al. (2017), including how they were estimated, the processes included, and the robustness. That paper built on RF values calculated by Bellouin et al. (2016). The paper applied four different coupled chemistry-climate models or CTMs. They compared control simulations with perturbed simulations where emissions were reduced by 20% for one species and one emission region. We apply the average values across models. For the aerosols and aerosol precursors, three out of four models included the aerosol direct and first indirect (cloud-albedo) effect. RF for BC deposition on snow and ice surfaces and the semi-direct effect was estimated in one of the models. For the ozone precursors (NOx, CO, and VOC) and CH4, RF is modelled for the aerosol direct effect and first indirect effects, short-lived ozone effect, methane effect, and methane-induced ozone effect. Nitrate aerosols are also considered based on results from one model.

The matrix of regional response coefficients (RCS), which enables us to go from regional RFs to regional temperature responses and ARTPs, are also presented in detail by Aamaas et al. (2017). The RCS values are mostly based on coefficients modelled by Shindell and Faluvegi (2009). A weakness with our chosen method is that Shindell and Faluvegi (2009) is to our knowledge the only study that provide the necessary relationships between regional RFs and regional temperatures to create RCS values."

We have also some more text in the paragraph about uncertainties in Section 2.3:

"Previous work by Aamaas et al. (2016) shows that the assumption of independent radiative forcing uncertainties gives a total uncertainty range for emission reductions for a mix of species that is similar to the range seen between different models. Further, they also found robustness for the method we use here to estimate temperature changes, such as models agreeing on whether different mitigation scenarios lead to warming or cooling."

P5 l18-l26 Please quote values +/- errors correctly. It is incorrect to quote -0.33 +/- 0.083 C. The error should probably only be quoted to one significant figure, although you may feel justified to quote to two, as you have done. But the value then needs to be quoted to the same number of decimal places as the error, i.e. it should be, e.g., -0.335 +/- 0.083 C, or -0.33 +/- 0.08 C. The same inconsistency

appears on several of the subsequent lines. We have edited how we quote the errors, mostly to one significant figure.

P6 l2 on -> in OK

P6 l5 on -> to OK

P7 l2 estimates OK

P7 l11 usage -> use OK

P7 l20 it contributes -> they contribute OK

P8 l14 values -> value OK

P8 l15 is -> are OK

P8 l24 SLCFs reduction -> SLCF reductions OK

P8 l30 implicitly OK

P9 l3 how -> what OK

P9 l18 …may be smaller than those estimated here… OK

Figure 2 caption should explain the origin of the error bars. We have added:
"Error bars representing 1 standard deviation are given for the net response in 2030, 2050, and 2100. They are calculated based on literature values for gaussian uncertainties in per-component RF, assuming no inter-species correlation, and estimated using a Monte Carlo analysis (100 000 pulls) where component forcing values are drawn from within the uncertainty distributions."

Figure 4 caption should explain the origin of the error bars (on the global values). We have added:
"Error bars representing 1 standard deviation are given for the sectors for the global temperature response. They are calculated based on literature values for gaussian uncertainties in per-component RF, assuming no inter-species correlation, and estimated using a Monte Carlo analysis (100 000 pulls) where component forcing values are drawn from within the uncertainty distributions."

Figure 5 caption – the explanation of the error bars could be clearer. We have edited to:
"Error bars representing 1 standard deviation are included. The blue and black error bars are calculated based on literature values for gaussian uncertainties in per-component RF, assuming no inter-species correlation, and estimated using a Monte Carlo analysis (100 000 pulls) where component forcing values are drawn from within the uncertainty distributions. The blue error bars indicate the uncertainty for the emission regions, the black error bars for the emission sectors. The grey error bars are estimated from uncertainty in the climate sensitivity based on Monte Carlo analysis (100 000 pulls) with values drawn from within the lognormal uncertainty distribution."

Reviewer 2:

In this study, the authors investigate the potential temperature implications of stringent air quality policies, by applying matrices of regional temperature responses to new pathways for future anthropogenic emissions of aerosols, methane (CH4) and other short-lived gases. This is an

interesting and relevant topic since there are still a lot of uncertainties on how regional temperatures are affected by ambitious SLCF emission mitigation policies.

General comments

The Introduction is too short, I suggest the authors to add more information about SLCF description. For example, here you show results for BC, OC, SO2, NOx, CO, VOC, and CH4. Some description about their cooling/warming impact of them will help to a better understand of the results. Also, maybe a bit more description of the ECLIPSE project would be good, since this works is strongly connected to it.

We have added several sentences on the ECLIPSE projects, its findings, and connections with our manuscript in the fourth paragraph of the introduction:

"That paper synthesized the work in the project ECLIPSE (Evaluating the Climate and Air Quality Impacts of Short-Lived Pollutants). The project designed realistic and effective mitigation scenarios for SLCFs and quantifying its climate and air quality impacts. The work started with producing new emission inventories for the recent past and until 2050. Those emissions were applied in several advanced Earth system models (ESMs) and chemistry transport models (CTMs). The climate impacts were estimated with two different paths of research, where the first was to calculate radiative forcing (RF) and then produce emission metrics such as ARTP. The second path was on modelling transient climate responses with ESMs. Results from the first path were applied in an integrated assessment model to identify emission mitigation measures that are both beneficial for air quality and short-term climate impact. That study found that estimates on global temperature change are similar for the decade 2041-2050 by applying these two different paths. Further, the two different research paths partly agree on how much emission changes in CH4 is responsible for the temperature change versus emission changes of the other SLCFs. Our study utilizes several aspects of the ECLIPSE research, including emission inventories, mitigation pathways, and ARTP values."

We have also added these sentences to the first paragraph of the introduction, to describe the different SLCFs in more detail:

"The SLCFs considered here are black carbon (BC), organic carbon (OC), sulphur dioxide (SO2), nitrogen oxides (NOx), carbon monoxide (CO), volatile organic compounds (VOC), and methane (CH4). CH4, which is a greenhouse gas and a precursor of O3 and stratospheric water vapor, is the SLCF that gives the largest warming at current emission levels. BC (also known as soot) is a result of incomplete combustion, that causes warming through absorption of sunlight and reduced albedo of contaminated snow and ice surfaces, but also cooling, mainly from affecting clouds. Removing all anthropogenic BC emissions would cause a cooling of -0.05 °C according to Stohl et al. (2015). Several aerosols are cooling the climate through scattering solar radiation and altering the radiative properties of clouds, with sulphate aerosol formed from SO2 and ammonia (NH3) giving the largest cooling. Stohl et al. (2015) estimate that removing all anthropogenic emissions of SO2 would increase the global temperature by 0.69 °C. OC is another cooling aerosol, of which a complete removal of anthropogenic OC emissions would lead to a warming of 0.13 °C (Stohl et al., 2015). The ozone-precursors NOx, CO, and VOC produce tropospheric O3, which is a greenhouse gas. Emissions of these species will also impact the hydroxyl radical (OH) concentration, which impacts CH4. The impact of current emissions of these ozone precursors is small compared to the impact of current emissions of CH4 and SO2."

Some figures are not well described in the text, for example, the authors directly mention Fig. 2 or 3 after describing a result obtained. The figures should be defined saying what it is representing there, and if needed, some explanation about how to interpret the graphic (if I got it right, the different symbols in fig 3 shows the influence of the different sectors of that region in a latitude band). This

way it will be easier to follow the text. Please, do it whenever the figure has not been described in advanced. There is also a lack of mention some figures that I will point out in the technical comments.

We have clearer presented the figures in the text. Further, we have added several callouts to the figures in the text. We have added a sentence for captions for Figure 3 and 4 that explains the symbols.

A better organization of the results must be done. It would be better to organize them in subsections, like "Global/Regional temperature change" or/and "Results by regions", as an example. Furthermore, more quantitative results could be added. Complementing with a table would be helpful for a better overview of the results found in this study (and comparing the results found in Stohl et al., 2015). We have added subtitles. These are Global temperature change, Regional temperature change, Temperature change by emission region, Temperature change by emission sector, Uncertainties. We have also added a table (Table 2), which is an overview of the results given in Figures 2, 3, and 4. Further, we have extended the result section to clearer organize the results and present the results in more details.

Specific comments

Page 1, line 16, authors state that "cutting CH4 and BC emissions contribute the most. This could offset warming equal to approximately 15 years of current global CO2 emissions." How do you get to this conclusion? I haven't seen it in the manuscript.

This point is discussed in the third paragraph of the discussion section. We have revised the sentence to make the point clearer:

"The net global cooling could offset warming equal to approximately 15 years of current global CO2 emissions."

Page 2, line 24, when mentioning the work of Stohl et al., 2015, although the authors mention it throughout the text, it will be good to have more description of the what they found. See reply to the first general comment by the Reviewer. We have expanded this paragraph to better present ECLIPSE, including some of their findings.

Page 2, line 29, here the description of the scenarios is done. I have several comments:

• The description of the SLCP_scen is not clear for me, so the difference is that the SLCP_scen only has 50 different mitigation measures on SLCFs compared to the 2 MTFR; a) how many has the MTRF scenario? And b) in what is it based to be called optimal? We have added one sentence regarding a): "The model behind includes more than 2000 technologies to control air pollutant emissions and 500 options to control greenhouse gas emissions." b) SLCP_scen is optimal in the sense of reducing the global temperature, hence, climate-optimal. This should be clearer with the new table (see point below) and modified description.

• It would be helpful to have the scenarios description as a list or as a table with a short description. We have produced a table (Table 1) as an overview of the three different scenarios. Some sentences from the text have been moved into the table.

• Check that you call the baseline scenario as "baseline" in the text. There are couple of times where you call it CLP, and sometimes is confusing to follow all the acronyms. We think the Reviewer is pointing out to CLE. We have either replaced CLE with baseline or added the word baseline in the text.

Page 3, line 7, IIASA has not been described. We have written out "The International Institute for Applied Systems Analysis".

Page 5, line 5, do you refer to the results shown in figure 5? If so, you could refer to it here. We refer to Figure 5 in the last section of the results section. We prefer not to link a figure from the results section to the method section.

Page 5, line 20, "The global temperature change is calculated as the area-weighted sum of the net regional changes given by equation 1." Can be added to the subsection 2.2, after describing the ARTP. We have moved this sentence to Section 2.3, as this section is most suitable for this information.

Page 5, line 28, "a warming of more tan 0.2ºC" how do you get this value? This finding is from Figure 2a, which we now have added a call-out to in the previous sentence. We have also revised the sentence to better reflect that this warming is solely from the reduction of the cooling components, by adding "from those".

Page 6, what about the warming temperature response found in fig. 3b and 4b? Authors only focus on cooling temperature response results. This comment could point to two different issues. First, that we don't discuss the warming responses. But we describe the net effects, which is a combination of the warming and cooling temperature responses. Second, and most likely point raised, that we don't present the findings in Figures 3b and 4b. We do write about these results and have added callouts to the b-figures to show this clearer.

Page 7, line 14, to what scenario does the value -0.33 ºC correspond? We have added "in SLCP_scen" in the sentence.

Technical comments

Page 1, line 13, "using existing regional temperature change potential (ARTP)" did you mean, using absolute existing regional temperature change potential? OK

Page 1, line 25, add a comma after "pollution". OK

Page 3, line 16, move "in Table 3 in Stohl et al. (2015) " to the beginning of the sentence in line 15 to avoid "Stohl et al. (2015). Stohl et al. (2015) " OK

Page 4, line 14, "CLE" to "baseline". OK

Page 5, line 25, add Fig. 2A somewhere in this line. OK

Page 5, line 30, add Fig. 2B somewhere in this line. OK

Page 6, line 9, "CLE" to "baseline". We have changed to "baseline CLE".

Page 6, line 14, in "mitigation scenarios" do the authors refer to MTRF?. If so:
Page 6, line 14, add Fig. 3b after "rest of the World". This sentence refers to both Figure 3A and 3B. We have added references to both Figure 3a and 3b in the paragraph.

Page 6, line 21, it should be Fig. 4a. OK

Page 6, line 25, add Fig. 3b after "to cooling.". OK

Page 10, line 11, doi is missing. But the DOI is there already: 10.1146/annurev-environ-052912-173303

**The regional temperature implications of strong air quality measures**

Borgar Aamaas[1], Terje K. Berntsen[1,2], Bjørn H. Samset[1]

[1]CICERO Center for International Climate Research, PB 1129 Blindern, 0318 Oslo, Norway
[2]Department of Geosciences, University of Oslo, PB 1047 Blindern, 0316 Oslo, Norway

5    *Correspondence to*: Borgar Aamaas (borgar.aamaas@cicero.oslo.no)

**Abstract.** Anthropogenic emissions of short-lived climate forcers (SLCFs) affect both air quality and climate. How much regional temperatures are affected by ambitious SLCF emission mitigation policies, is ,however, still uncertain. We investigate the potential temperature implications of stringent air quality policies, by applying matrices of regional temperature responses to new pathways for future anthropogenic emissions of aerosols, methane ($CH_4$) and other short-lived

10   gases. These measures have only minor impact on $CO_2$ emissions. Two main options are explored, one with climate optimal reductions (i.e. constructed to yield a maximum global cooling) and one with maximum technically feasible reductions. The temperature response is calculated for four latitude response bands (90-28° S, 28° S-28° N, 28-60° N, and 60-90° N) by using existing Absolute Regional temperature change potential (ARTP) values for four emission regions: Europe, East Asia, shipping, and the rest of the world. By 2050, we find that global surface temperature can be reduced by -0.3±0.08 °C

15   with climate-optimal mitigation of SLCFs relative to a baseline scenario, and as much as -0.7 °C in the Arctic. Cutting $CH_4$ and BC emissions contribute the most. This could offset warming equal to approximately 15 years of current global $CO_2$ emissions. On the other hand, mitigation of other SLCFS (e.g., $SO_2$) leads to warming. If SLCFs are mitigated heavily, we find a net warming of about 0.1 °C, but when uncertainties are included a slight cooling is also possible. In the climate optimal scenario, the largest contributions to cooling comes from the energy, domestic, waste, and

20   transportation sectors. In the maximum technically feasible mitigation scenario, emission changes from the sectors industry, energy, and shipping will give warming. Some measures, such as in the sectors agriculture waste burning, domestic, transport, and industry, have  large impact on the Arctic, especially by cutting BC emissions in winter in areas near the Arctic.

**1 Introduction**

25   Poor air quality is an issue of global concern, with health and welfare impacts affecting billions of people (WHO, 2016;Dockery et al., 1993;Di et al., 2017). Additionally, many of the components that make up air pollution, also lead to radiative forcing impacting climate, through scattering or absorbing solar radiation, or by acting as greenhouse gases (Myhre et al., 2013b;von Schneidemesser et al., 2015). The net and individual climate impacts of present emissions of such short-lived climate forcers (SLCFs) have been extensively studied, but are however still poorly constrained (Stohl et al.,

30   2015;Aamaas et al., 2016;Myhre et al., 2017;Samset et al., 2018). The SLCFs considered here are black carbon (BC),

organic carbon (OC), sulphur dioxide ($SO_2$), nitrogen oxides ($NO_x$), carbon monoxide (CO), volatile organic compounds (VOC), and methane ($CH_4$). $CH_4$, which is a greenhouse gas and a precursor of $O_3$ and stratospheric water vapor, is the SLCF that gives the largest warming at current emission levels. BC (also known as soot) is a result of incomplete combustion, that causes warming through absorption of sunlight and reduced albedo of contaminated snow and ice surfaces,

5   but also cooling, mainly from affecting clouds. Removing all anthropogenic BC emissions would cause a cooling of -0.05 °C according to Stohl et al. (2015). Several aerosols are cooling the climate through scattering solar radiation and altering the radiative properties of clouds, with sulphate aerosol formed from $SO_2$ and ammonia ($NH_3$) giving the largest cooling. Stohl et al. (2015) estimate that removing all anthropogenic emissions of $SO_2$ would increase the global temperature by 0.69 °C. OC is another cooling aerosol, of which a complete removal of anthropogenic OC emissions would lead to a warming of

10   0.13 °C (Stohl et al., 2015). The ozone-precursors $NO_x$, CO, and VOC produce tropospheric $O_3$, which is a greenhouse gas. Emissions of these species will also impact the hydroxyl radical (OH) concentration, which impacts $CH_4$. The impact of current emissions of these ozone precursors is small compared to the impact of current emissions of $CH_4$ and $SO_2$.

In the coming decades, mitigation of $CO_2$ and other long-lived greenhouse gases (LLGHGs) is vital for the success of the

15   goals in the Paris Agreement (UNEP, 2016). Concurrently, we expect large changes in SLCF emissions, in response to air quality policies, additional climate change mitigation efforts, and due to co-emissions with LLGHGs. As some SLCFs cool the climate, others warm it, and some may do both at different times after emission, the exact mitigation pathways of SLCFs will be of importance for the near-term rate and magnitude of warming – both globally and regionally. While several studies have analysed the impact on $CO_2$ mitigation of SLCFs (e.g., Rogelj et al., 2014), our study does not consider $CO_2$ emissions,

20   but investigates a set of air quality measures that mainly influence SLCFs emissions.

Designing mitigation measures with both air quality and climate change in mind is however not straightforward, as warming SLCFs are often co-emitted with cooling SLCFs. Some authors have argued that a mitigation focus on SLCFs can be counterproductive, as this may lead to relaxing efforts on reducing $CO_2$ emissions (Pierrehumbert, 2014;Shoemaker et al.,

25   2013). However, if this is done in a consistent way using emission metrics with appropriate time-horizons, this can be avoided (Berntsen et al., 2010). Another argument against SLCF mitigation today is that the long-term cooling potential of emission reductions is limited, and that delaying mitigation of SLCFs has only minor impact on temperature stabilization and peaking in the future (e.g., Pierrehumbert, 2014). However, SLCF mitigation is already occurring as part of air quality policy (Li et al., 2017) and is expected to continue in the coming decades regardless of the level of climate mitigation ambitions

30   (Victor et al., 2015;Rao et al., 2017). Stohl et al. (2015) showed (applying the Absolute Regional Temperature change Potential (ARTP) methodology) that climate optimal reductions of SLCFs, i.e. the combination of measures which maximize temperature reduction, may lower the global temperature by 0.22 °C in 2041-2050 compared to a reference scenario. In comparison, a complete removal of anthropogenic emissions of black carbon (BC), organic carbon (OC) and $SO_2$ (sulphate aerosol precursor)aerosol emissions (BC, OC and $SO_2$) would induce a global mean surface heating of 0.5–1.1°C, according

to four recent climate models (Samset et al., 2018). Going beyond temperature and precipitation impacts, SLCF emission mitigation is also known to have multiple co-benefits and trade-offsclose links with a range of the UN Sustainable Development Goals (SDGs) (Haines et al., 2017). The co-benefits are generally larger than the trade-offs. Among the most well-known co-benefits, we find that SLCF mitigation will reduce air pollution and, hence, reduce premature deaths (SDG3), as well as reduce crop losses (SDG2).

Recently, Stohl et al. (2015) gave a general overview of the temperature reduction potential of SLCFs. That paper synthesized the work in the project ECLIPSE (Evaluating the Climate and Air Quality Impacts of Short-Lived Pollutants). The project designed realistic and effective mitigation scenarios for SLCFs and quantifying its climate and air quality impacts. The work started with producing new emission inventories for the recent past and until 2050. Those emissions were applied in several advanced Earth system models (ESMs) and chemistry transport models (CTMs). The climate impacts were estimated with two different paths of research, where the first was to calculate radiative forcing (RF) and then produce emission metrics such as ARTP. The second path was on modelling transient climate responses with ESMs. Results from the first path were applied in an integrated assessment model to identify emission mitigation measures that are both beneficial for air quality and short-term climate impact. That study found that estimates on global temperature change are similar for the decade 2041-2050 by applying these two different paths. Further, the two different research paths partly agree on how much emission changes in CH4 is responsible for the temperature change versus emission changes of the other SLCFs. Our study utilizes several aspects of the ECLIPSE research, including emission inventories, mitigation pathways, and ARTP values. WeOur study explores these findings in Stohl et al. (2015) further for individual emission regions and emission sectors using updated data and methods, following pathways that focus on air quality concerns. Our focus is on the temperature effects of SLCFs, however, mitigation of these components can also help to achieve several of the Sustainable Development Goals (Shindell et al., 2017).

A detailed look into what sectors and regions that contribute to the mitigation potential from SLCF reductions requires a comprehensive emission dataset. As part of the ECLIPSE project, emission inventories and scenarios for future emissions (for the period 1990-2050) of SLCFs were produced (Klimont et al., 2017;Klimont et al., In prep.). The scenarios describe three different futures with different mitigating ambitions (see Table 1), one baseline with current legislation (CLE), one with as much mitigation of SLCFs as possible (MTFR), and one which only include measures that lead to net global cooling. 
[revised manuscript text omitted]
 ARTP dataset utilized here are presented in detail by Aamaas et al. (2017), including how they were estimated, the processes included, and the robustness. That paper built on RF values calculated by Bellouin et al. (2016). The paper applied four different coupled chemistry-climate models or CTMs. They compared control simulations with perturbed simulations where emissions were reduced by 20% for one species and one emission region. We apply the average values across models. For the aerosols and aerosol precursors, three out of four models included the aerosol direct and first indirect (cloud-albedo) effect. RF for BC deposition on snow and ice surfaces and the semi-direct effect was estimated in one of the models. For the ozone precursors (NOx, CO, and VOC) and $CH_4$, RF is modelled for the aerosol direct effect and first indirect effects, short-lived ozone effect, methane effect, and methane-induced ozone effect. Nitrate aerosols are also considered based on results from one model.

The matrix of regional response coefficients (RCS), which enables us to go from regional RFs to regional temperature responses and ARTPs, are also presented in detail by Aamaas et al. (2017). The RCS values are mostly based on coefficients modelled by Shindell and Faluvegi (2009). A weakness with our chosen method is that Shindell and Faluvegi (2009) is to our knowledge the only study that provide the necessary relationships between regional RFs and regional temperatures to create RCS values.

Uncertainties (1 standard deviation) in the global temperature response have been estimated given a Monte Carlo analysis of 100 000 simulations. This analysis is based on a probability density function defined by model based estimates of uncertainties in direct radiative forcing from the literature (Myhre et al., 2013b;Myhre et al., 2013a) with the same treatment of radiative forcing uncertainty as in Lund et al. (2017) (see also the Supplement). Radiative forcing from each species is treated as a random variable. The distribution for the total uncertainty is derived by summing the probability density functions of all species. We assume that the radiative forcing uncertainties are independent in these calculations. Previous work by Aamaas et al. (2016) shows that the assumption of independent radiative forcing uncertainties gives a total uncertainty range for emission reductions for a mix of species that is similar to the range seen between different models. Further, they also found robustness for the method we use here to estimate temperature changes, such as models agreeing on whether different mitigation scenarios lead to warming or cooling. Also, note that the multi-model studies used as input were run with unified emissions. This particularly affects BC, where the current substantial uncertainty in annual emissions (Bond et al., 2013;Cohen and Wang, 2014) will not be represented. We compare our derived uncertainties to the influence of low and high climate sensitivities in the literature, 1.5 and 4.5 °C for a doubling of $CO_2$ (Bindoff et al., 2013). Here, we adopt a lognormal distribution and assume the value range covers 1 standard deviation. Uncertainties are not given for the latitude bands as a formal quantification of uncertainties for the ARTPs has not been produced.

**3 Results**

As our analysis can be viewed from multiple dimensions, we present the results by focusing on one and one dimension. An overview of our temperature response estimates relative to the baseline is given in Table 2, which are presented in detail in the following sections. Temperature effects in 2050 are presented for the global and regional level, for species, for emission regions, and for emission sectors.

**3.1 Global temperature change**

Figure 2 shows the temporal temperature response from 2010 until 2100 with the two scenarios relative to the baseline for the different species and the net response. As for the following figures, results for SLCP $_{scen}$ relative to the baseline are found in the upper panel and MTFR relative to the baseline in the lower panel. If SLCFs are mitigated in a climate-optimal manner, we estimate a maximum change in global temperature of -0.3±0.01 °C by 2050, relative to current legislation, increasing to about -0.4 °C later in the century ending at -0.4±0.1 °C in 2100 (see Fig. 2a, black line, and Table 2).  The temperature response of aggressive mitigation of SLCFs (MTFR) leads gradually to a small change in temperature  of 0.01±0.11 °C in 2050 relative to the baseline, which seems to be counter-productive in terms of goals limiting the global temperature increase (see Fig. 2b). As the uncertainty interval is large, since large emission cuts of warming and cooling components cancel each other almost out (about 0.7 °C cooling and warming in 2050, see Fig. S3 in the Supplement for cooling and warming separated for both MTFR and SLCP $_{scen}$), we cannot rule out that this scenario may lead to cooling. In the climate-optimal scenario (SLCP $_{scen}$), CH $_4$ (-0.21±0.02 °C in 2050 and increasing in magnitude) and BC (-0.2±0.07 °C in 2050) are the main drivers of the temperature reductions (Fig. 2a). The measures will also reduce co-emissions of cooling species causing a warming from those of more than 0.2 °C in 2050. The main warming contributions are emission reductions of OC and NO $_x$, with small impacts from other SLCFs. The main difference to the maximum reduction scenario (MTFR) is the large warming contribution for MTFR (0.4±0.1 °C in 2050) from SO $_2$ reductions, as well as additional warming from NO $_x$ reductions (Fig. 2b).

**3.2 Regional temperature change**

The temperature responses in the four latitudinal bands are given in Fig. 3 for different emission regions and emission sectors, all responses are relative to the baseline. The global responses are found to the right, while the responses in the latitude bands from south to north are given from left towards right. The symbols are the net response for each emission region. In this section, we discuss differences in the response between the latitude bands. The Arctic (60-90° N) is the region that is the most sensitive to the mitigation scenarios for all emission regions (see Fig. 3 and Table 2), followed by northern mid-latitudes (28-60° N), as the climate sensitivities are largest for those regions and most of the emissions occur in the Northern Hemisphere (Aamaas et al., 2017). In SLCP $_{scen}$, the cooling in the Arctic (-0.7 °C in 2050) is more than twice

the global average. This sensitivity in the Arctic is larger than for reductions of $CO_2$, which would be roughly 50% when applying the ARTP concept to $CO_2$. This amplification in the Arctic is larger than the average for mitigation of European emissions and smaller for mitigation of East Asian emissions (see Fig. 3a). Measures on BC emissions during winter in the Northern Hemisphere contribute to this amplification. In terms of sectors, mitigation measures on SLCFs from agriculture waste burning, domestic, transportation, and industry have larger than average influence on the Arctic relative to the global average (Fig. 4). Some variability is also seen for the Arctic. While MTFR will lead to warming globally relative to baseline CLE, a cooling of the same magnitude is estimated for the Arctic (see Fig. 3b). The net cooling in the Arctic is driven by emissions from rest of the World, while mitigation in the shipping sector leads to warming for both and the net effect of European mitigation is near zero .

**3.3 Temperature change by emission region**

The emission region that contributes the most in the mitigation scenarios is Rest of the World (see Fig. 3 and Table 2). In the Supplement, we indicate that rest of Asia and other developing regions are the most important regions (as seen in Stohl et al., 2015), although our ARTP dataset limits us from making clear conclusions of what sub regions have the largest cooling potential. In SLCP$_{scen}$, mitigation leads to cooling from all emission regions and emission sectors except global shipping (Fig. 3a). In MTFR, warming globally is estimated for rest of the world and shipping, while near zero change for Europe and a cooling contribution for East Asia (Fig. 3b).

**3.4 Temperature change by emission sector**

In Fig. 4, the temperature responses in the four latitudinal bands are given for different emission sectors and separated by the emitted species, all responses are relative to the baseline. 
[revised manuscript text omitted]

15   Table 1: An overview of the three emission scenarios with different mitigating ambitions investigated in this study. The emission inventories and scenarios for the period 1990-2050 have been produced by Klimont et al. (2017);Klimont et al. (In prep.);Stohl et al. (2015).

| Scenario | Acronym | Description and mitigation measures |
|---|---|---|
| Baseline – current legislation | CLE | The baseline scenario assumes implementation of current (2015) legislation. Both current and planned environmental laws are included while considering known delays, but assuming full enforcement in the future. |
| Mitigation - maximum technically feasible reductions | MTFR | The most ambitious mitigation scenario, where SLCFs are cut as much as possible (although without changes in consumer behaviour, structural changes in transport, agriculture or energy supply or additional climate policies) due to air quality concerns. This is a very policy demanding scenario, as most emissions are reduced by 60-80 % within a few decades. The model behind includes more than 2000 technologies to control air pollutant emissions and 500 options to control greenhouse gas emissions. |
| Mitigation - climate-optimal mitigation scenario | SLCP$_{scen}$ | A subset of MTFR containing about 50 different mitigation measures on SLCFs. Only measures that are estimated to lead to net global cooling, while reduction of co-emitted cooling species are accounted for, are included, hence, climate-optimal. These measures are technical measures on emissions of CH4 and BC, as well as non-technical measures to eliminate BC. |

Table 2: The global and regional temperature responses in 2050 for SLCP$_{scen}$ and MTFR scenarios relative to the baseline
20   CLE. Global temperature responses are given for the net, as well as for all the species, emission regions, and emission sectors at a global level. In the lower part, temperature responses in the four latitude bands are shown for global emissions. This table is a synthesis of Fig. 2, 3, and 4. The sectors included are agriculture (agr), agriculture waste burning (awb), domestic (dom), energy (ene), industry (ind), solvent (slv), transportation (tra), waste (wst), and shipping (shp).

| ΔT [°C] in 2050 | SLCP$_{scen}$ - CLE | MTFR - CLE |
|---|---|---|
| Sum | -0.3 | 0.1 |
| Species | SLCP$_{scen}$ - CLE | MTFR - CLE |
| BC | -0.2 | -0.2 |
| OC | 0.1 | 0.1 |
| SO2 | 0.002 | 0.4 |
| NOx | 0.02 | 0.04 |
| CO | -0.03 | -0.03 |
| VOC | -0.02 | -0.02 |
| CH4 | -0.2 | -0.2 |
| Regions | SLCP$_{scen}$ - CLE | MTFR - CLE |
| EUR | -0.01 | 0.003 |
| EAS | -0.06 | -0.03 |
| ROW | -0.3 | 0.05 |
| SHP | 0.002 | 0.03 |
| Sectors | SLCP$_{scen}$ - CLE | MTFR - CLE |
| dom | -0.06 | -0.05 |
| ene | -0.1 | 0.05 |
| ind | -0.02 | 0.1 |
| tra | -0.04 | -0.03 |
| wst | -0.06 | -0.06 |
| awb | -0.004 | -0.004 |
| shp | 0.002 | 0.03 |
| agr | -0.01 | -0.01 |
| slv | -0.006 | -0.005 |
| ΔT in latitude bands | SLCP$_{scen}$ - CLE | MTFR - CLE |
| 90° S-28° S | -0.2 | -0.02 |
| 28° S-28° N | -0.3 | 0.1 |
| 28° N-60° N | -0.5 | 0.1 |
| 60° N-90° N | -0.7 | -0.02 |

**Figure 1:** The global emission levels relative to the 1990 level for  CLE (a), SLCP_scen (b) and MTFR (c). The 1990 emission level for each SLCF is normalized to 100.

**Figure 2:** Global temperature response due to the SLCP_scen (A) and MTFR (B) scenarios  relative to the baseline CLE scenario. Future global temperature change will also be impacted by historic and baseline emissions, which are not accounted for here. Error bars representing 1 standard deviation are given for the net response in 2030, 2050, and 2100. They are calculated based on literature values for gaussian uncertainties in per-component RF, assuming no inter-species correlation, and estimated using a Monte Carlo analysis (100 000 pulls) where component forcing values are drawn from within the uncertainty distributions.

**Figure 3:** The temperature response in the latitude bands and globally in 2050 for emission regions and emission sectors for SLCP_scen (a) and MTFR (B) scenarios relative to the baseline CLE. The emission regions are Europe (EUR), East Asia (EAS), global shipping (SHP), and the rest of the World (ROW). The net response in the latitude bands due to emissions from each emission region is given by the symbols. The emission sectors are agriculture (agr), agriculture waste burning (awb), domestic (dom), energy (ene), industry (ind), solvent (slv), transportation (tra), waste (wst), and shipping (shp).

**Figure 4:** The temperature response in the latitude bands and globally in 2050 for emission sectors and species for SLCP_scen (a) and MTFR scenario (b) relative to the baseline CLE. Future global temperature change will also be impacted by historic and baseline emissions, which is not accounted for here. The emission sectors are agriculture (agr), agriculture waste burning (awb), domestic (dom), energy (ene), industry (ind), solvent (slv), transportation (tra), waste (wst), and shipping (shp). The net response in the latitude bands due to emissions from each emission sector is given by the symbols.

Error bars representing 1 standard deviation are given for the sectors for the global temperature response. They are calculated based on literature values for gaussian uncertainties in per-component RF, assuming no inter-species correlation, and estimated using a Monte Carlo analysis (100 000 pulls) where component forcing values are drawn from within the uncertainty distributions.

**Figure 5:** The global temperature response in 2050 in mitigation scenarios relative to the baseline for emission regions and emission sectors for SLCP_scen and MTFR scenario. Error bars representing 1 standard deviation are included. The blue and black error bars are calculated based on literature values for gaussian uncertainties in per-component RF, assuming no inter-species correlation, and estimated using a Monte Carlo analysis (100 000 pulls) where component forcing values are drawn from within the uncertainty distributions. The blue error bars indicate the uncertainty for the emission regions, the black error bars for the emission sectors. The grey error bars are estimated from uncertainty in the climate sensitivity based on Monte Carlo analysis (100 000 pulls) with values drawn from within the lognormal uncertainty distribution.

[Figure]

[Figure]

**A) Global temperature response difference for SLCP$_{scen}$ - CLE**

[Figure]

**B) Global temperature response difference for MTFR - CLE**

[Figure]

**A) Temperature response in 2050, SLCP$_{scen}$ - CLE**

[Figure]

**B) Temperature response in 2050, MTFR - CLE**